# Sense of doubt: inaccurate and alternate locations of virtual magnetic displacements may give a distorted view of animal magnetoreception ability

Will T. Schneider [1]✉, Florian Packmor[2], Oliver Lindecke[3,4] & Richard A. Holland[1,4]

Virtual magnetic displacements are used to examine the magnetoreceptive ability of animals by changing the local magnetic field to emulate one that exists elsewhere. This technique can be used to test whether animals use a magnetic map. The viability of a magnetic map is dependant upon which magnetic parameters an animal's coordinate system is composed of, and how sensitive they are to those parameters. Previous research has not considered the degree to which sensitivity can change an animal's impression of where a virtual magnetic displacement is located. We re-assessed all published studies that use virtual magnetic displacements assuming the highest likely level of sensitivity to magnetic parameters in animals. The vast majority are susceptible to the existence of alternate possible virtual locations. In some cases, this can cause results to become ambiguous. We present a tool for visualising all possible virtual magnetic displacement alternative locations (ViMDAL) and propose changes to how further research on animal magnetoreception is conducted and reported.

[1] School of Natural Sciences, Bangor University, Bangor, Gwynedd LL57 2UW, UK. [2] Lower Saxon Wadden Sea National Park Authority, 26382 Wilhelmshaven, Germany. [3] Institute of Biology and Environmental Sciences, University Oldenburg, 26111 Oldenburg, Germany. [4] These authors jointly supervised this work: Oliver Lindecke, Richard A. Holland. ✉email: w.schneider@bangor.ac.uk

For more than two decades, virtual magnetic displacements have been used to manipulate animal behaviour and study it as a reaction to the Earth's magnetic field[1–3]. This method allows changes to the magnetic field in a confined space, which may emulate conditions that exist elsewhere on Earth, or even conditions that are not found anywhere on the planet. By placing an animal inside magnetic field conditions that exist elsewhere, it then becomes 'virtually displaced' without having to physically transport it to the respective, magnetically simulated location. In this way, all other environmental cues of the actual location (e.g. field site or laboratory) remain unaltered, and any behavioural or physiological effects can be attributed to the changed magnetic field components alone. However, there are some caveats to the power of this technique. While advancements are being made in understanding the underlying physiological mechanisms that can allow animals to detect components of the Earth's magnetic field[4,5], there is a lack of knowledge of how this scales up in vivo. Many virtual magnetic displacement studies have been carried out on a wide range of taxa without clear knowledge of how sensitive the subject animals were to magnetic field parameters, or even which parameters they may be able to perceive at all. Depending on how sensitive an animal is to magnetic parameters, it may be possible to perceive a virtual magnetic displacement as existing at many alternate locations[6]. This is problematic because it is often a key requirement of interpretation that a virtually displaced individual will perceive its location to have changed to a specific location matching the new magnetic field conditions. Therefore, a lack of knowledge of the animals' sensitivity is potentially problematic and may lead to a misinterpretation of results from virtual magnetic displacement studies. In this review, we show that even when assuming extremely high levels of sensitivity, the locations matching the parameters of a virtual magnetic displacement are rarely unique. In some cases, this means that questions are posed that cannot be answered, leading to uncertainty in the interpretation of results. We do this by re-assessing virtual magnetic displacement studies using a modelling tool for visualising virtual magnetic displacement alternate locations (ViMDAL). This tool was developed in MATLAB and we provide it open source alongside this article.

To conduct a virtual magnetic displacement, a 3D magnetic coil system (e.g. Helmholtz or Merritt coil system) is used to alter the Earth's magnetic field. A three-axis coil system can alter the field strength in each of the $x$, $y$, and $z$ components. The $x$ and $y$ components change the horizontal orientation of the field and consequently define the direction of magnetic North. Therefore, in experiments that aim to test whether an animal has a magnetic compass, only the $x$ and $y$ components need to be modified (assuming a perfectly levelled coil)[7]. Additionally, experiments can test to see whether animals have a magnetic map. Magnetic maps are generally considered to be composed of up to three magnetic parameters; magnetic total intensity, magnetic inclination, and (less commonly) magnetic declination[8]. It is one or more of these three parameters (Fig. 1), rather than the raw $x$, $y$, and $z$ components of the field, that are generally considered to form an animal's magnetic map[9–11]. The feasibility of a magnetic map for navigation depends upon which of these parameters are detectable, and the location on Earth in which the animal needs to navigate[6,12]. There are vast areas of the Earth's surface in which the similar gradients of change of magnetic inclination and total intensity should not permit effective navigation, such as the Atlantic Ocean[6]. Despite this, many studies have obtained results suggestive of an ability to use a magnetic map within these problematic areas[3].

Whilst virtual magnetic displacements are a powerful tool for testing a magnetic sense, they also bear the potential of producing inconclusive results and rebuttable interpretations if critical a priori knowledge is missing. Prior to a virtual displacement study being carried out, there may be little knowledge of which magnetic parameters an animal can sense. Virtual magnetic displacements might then be used to determine if and how sensitive an animal is to which particular magnetic parameters. If an animal does not show a behavioural shift when a parameter is changed, then it may be decided that that parameter is not within its suite of possible cues. This is problematic and could lead to results that can be either false positive or negative. It may falsely appear that the animal is not sensitive to a magnetic parameter if it is not presented in a context that is ecologically relevant to the animal. The necessary ecological context to provoke behavioural change is very difficult to predict if there is already a lack of a priori knowledge of how or when these cues may be used. Conversely, animals may show behavioural changes that suggest sensitivity to a magnetic parameter, or even that the parameter is being used as a navigational cue, when instead it may be part of a more-complex behavioural response to feeling lost, or a non-specific effect to the treatment[13].

This article will focus on the lack of knowledge of exactly how sensitive an animal is to the three magnetic parameters termed total intensity, inclination, and declination. In fact, declination is rarely included as a possible cue in virtual magnetic displacement experiments, and so the basis of most of the research relies upon the usage of only two magnetic cues; total intensity and inclination[9]. Herein lies the key issue: total intensity and magnetic inclination have remarkably similar gradients of change across large parts of the globe. Both generally change along an approximate latitudinal axis and thus provide information that is not different enough from each other to drastically improve localisation ability[6]. This makes the lack of knowledge of how sensitive an animal is to these parameters a critical problem, because to realistically use inclination and total intensity as a bi-coordinate map, an incredibly high level of sensitivity would be required. We hand-pick examples below to show that even a generous assumption of very high sensitivities to magnetic fields can cause results of influential virtual magnetic displacement studies to become somewhat ambiguous.

## Results and discussion

**A tri-coordinate map.** Combining magnetic total intensity, inclination, and declination, would produce a tri-coordinate map that is less susceptible to issues of sensitivity than a bi-coordinate map of only intensity and inclination. As an example, we look at a study on the virtual magnetic displacement of reed warblers[14]. In this study, Kishkinev et al. tested the orientation of reed warblers under alternative magnetic conditions. Here we focus on two of those conditions. The first is the natural magnetic field of the test site in Austria, which is on the migratory route of the warblers. The second is a virtual magnetic displacement. Birds were not physically moved, but instead placed in an altered magnetic field. For this, the magnetic field parameters total intensity, inclination, and declination, were all changed to match those found several hundreds of kilometres Northeast. Using our tool to visually examine the parameters on a map, we see that the value of total intensity found naturally at the test site (48,500 nT) also exists at locations spanning longitudinally East and West (Fig. 2a, blue area). Total intensity for the virtual magnetic displacement was increased to 55,110 nT (Fig. 2a, pink area). The possible locations with this value of intensity do not follow a straightforward longitudinal pattern, instead they have a shifting gradient of change (Fig. 2a, pink area). For both the natural inclination value, and the inclination value of the virtual magnetic displacement, other possible locations with the same values span longitudinally (Fig. 2b). The declination isolines are not linear, though follow a

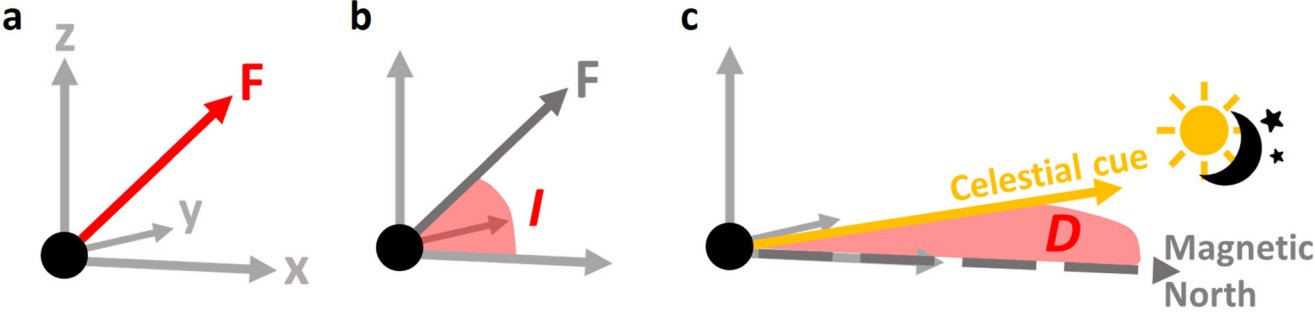

**Fig. 1 How are the magnetic parameters defined and how might animals sense them? a** Total Intensity, F. The total strength of the magnetic field at a particular location, combining the x, y, and z vectors. The magnetite hypothesis provides a physiological explanation for how animals might sense total intensity[10]. This theory suggests that animals may be able to sense the direction of the field. This enables the possibility for intensity to be used as both a magnetic compass and a map. **b** Inclination, I. The angle between the total intensity vector and the horizontal $x/y$ plane. Inclination ranges from 90° (at the North pole), through 0° (at the equator), to −90° (at the South pole). The radical pair hypothesis suggests a physiological mechanism for the detection of magnetic inclination[11]. It is theorised that inclination could be used both as a magnetic compass and in a magnetic map. **c** Declination. D The technical definition of declination is the angle between magnetic North and geographic (true) North. However, declination in the context of animal magnetoreception can be thought of as the angular difference between a magnetic orientation (here shown as North) and a celestial cue (as this is the only available proxy for geographic North). Any celestial landmark could be used. Because of this, it is important when carrying out declination exposure to make sure that the animal has been exposed to every possible way of determining their declination. A magnetic compass sense is required to determine declination, which could then be used as part of a map.

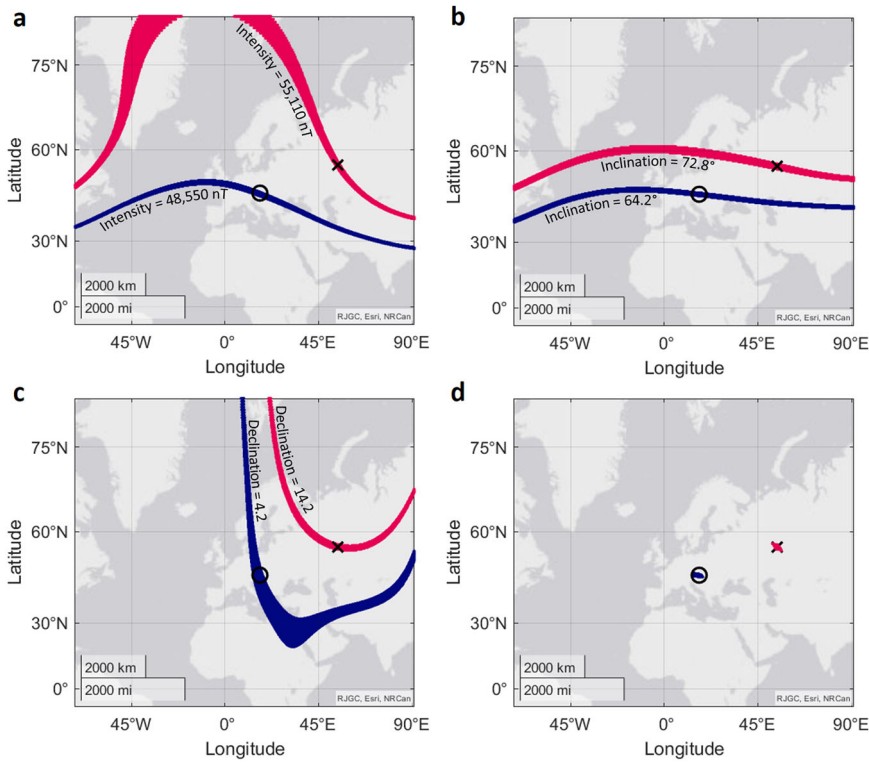

**Fig. 2 Possible locations with the same values of total intensity, inclination, declination, and all three parameters combined, as found at the natural site in ref. [14]. a** Locations matching the values (±200 nT) of total intensity found at the natural site (black cross, blue area), and the virtual magnetic displacement (black cross, pink area). **b** Locations matching the values (±0.5°) of inclination found at the natural site (black circle, blue area), and the virtual magnetic displacement (black cross, pink area). **c** Locations matching the values (±0.5°) of declination found at the natural site (black circle, blue area), and the virtual magnetic displacement (black cross, pink area). **d** Possible locations for the natural site (black circle, blue area) and the virtual magnetic displacement (black cross, pink area), when all three magnetic parameters are combined.

similar pattern for both the natural field value and the virtual magnetic displacement value (Fig. 2c). The different gradients of change of these three parameters makes their usefulness as part of a map dependent upon the context in which they are required. The ecological context is key to understanding how an animal may interpret such changes in magnetic field conditions. From a purely theoretical standpoint, however, to be able to pinpoint a single location on the Earth, then in this scenario, all three parameters must be combined (Fig. 2d). Therefore, despite the gradients of change being variable and unintuitive, theoretically it should be possible for a tri-coordinate magnetic map to allow an animal to localise their position to the exact location that is suggested in the paper. As we show in further examples, this is not always the case when instead using a bi-coordinate map.

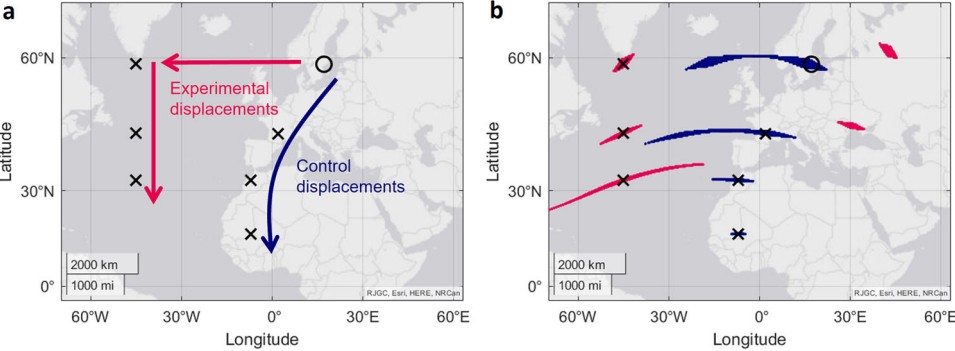

**Fig. 3 Bi-coordinate virtual magnetic displacements in ref. [16]. a** Experimental design of the study. The test site near where the birds were captured is marked with a black circle. The suggested virtual magnetic displacement locations are marked with a black cross. **b** All locations with the same values of total intensity and inclination as those of the virtual magnetic displacements (±200 nT and 0.5° inclination). Blue areas are the possible locations for the control displacement, and pink areas for the experimental displacements.

**Bi-coordinate maps in Europe.** The vast majority of virtual magnetic displacement studies do not provide animals with declination as a usable magnetic cue. Usually this is because animals are tested indoors without access to any reference cues needed to calculate declination (i.e. celestial cues). This leaves only total intensity and inclination. As mentioned above, and has been noted in other studies[6,12,15], the similarity in the gradients of change of these two parameters limits their viability as a magnetic map. Surprisingly, this has not prevented a great number of studies suggesting that virtual magnetic displacements using only intensity and inclination can be pinpointed on a map as a single location. To show how this can cause problems, we have selected a range of studies that have performed bi-coordinate virtual magnetic displacements. First, we look at a study on the fuelling behaviour of wheatears on the migration from Northern Europe to Western Africa[16]. After a period in the natural magnetic field in North Sweden (circle, Fig. 3a), birds were given a series of virtual magnetic displacements over the course of 20 days. These virtual locations either followed their normal migratory route (blue, 'control displacements', Fig. 3a), or a path South of Greenland (pink, 'experimental displacements', Fig. 3a). However, these are not the only locations where the combinations of totally intensity and inclination parameters exist. In fact, for every virtual magnetic displacement in this study, there are several other possible locations with the same values of inclination and intensity (±200 nT and 0.5° inclination, Fig. 3b). Within the region of interest, possible locations for the control displacements stretch into the Atlantic Ocean (Fig. 3b, blue areas). Possible locations for the experimental displacements exist in Eastern Europe, and for the final southernmost virtual magnetic displacement, they span a wide area of the Atlantic Ocean (Fig. 3b, pink areas). It may be that the alternate locations for the control displacements, stretching into the Atlantic Ocean, are not problematic because they are unlikely to be confused by the birds with the locations existing on their migratory route on land. However, if the same argument is to be applied to the idealised experimental displacement locations, then why might a bird predict their location to be even further outside of the normal range? Furthermore, possible locations for the virtual magnetic displacements that exist in Eastern Europe are not only closer to the test and capture site, but also are on land and far more likely to be within the birds' normal range. It therefore seems more probable that the experimental birds would perceive their location to be Eastwards rather than Westwards for the first two virtual magnetic displacements. Regardless, the birds in the two treatments did differ in their fuelling behaviour, thus there is no question that the magnetic field changes have had an effect. It is

unavoidable, however, that the interpretation of these results become less precise when considering the fact that the virtual magnetic displacement locations are not unique.

**Bi-coordinate maps in the Atlantic Ocean.** The Atlantic Ocean has been highlighted as a particular problem area for the bi-coordinate map[6]. In some areas, the isolines for inclination and total intensity follow paths that are near-identical. Despite knowledge of this challenge during the emergence of the magnetoreception field[17], many studies have since argued that animals are capable of using a bi-coordinate map in the area. In one of those, hatchling turtles were virtually displaced to four locations in the Atlantic Ocean[18]. Their orientation during each virtual magnetic displacement was examined, and it was found that their direction aligned with that of the Atlantic gyre. Again, we now consider a realistic sensitivity to the magnetic parameters and view the possible locations for the virtual magnetic displacements using our visualisation tool (Fig. 4). For virtual displacement (i), in the Mid-Atlantic, possible locations span a wide area of the Atlantic Ocean, and also appear in Southern Europe (Fig. 4, blue dots). We may reasonably exclude the locations in Southern Europe as they are on land and far from the migratory route. Considering only the locations in the Mid Atlantic, and that the orientation of the turtles for this virtual displacement was Eastwards, we may ask if this orientation is meaningful at each of

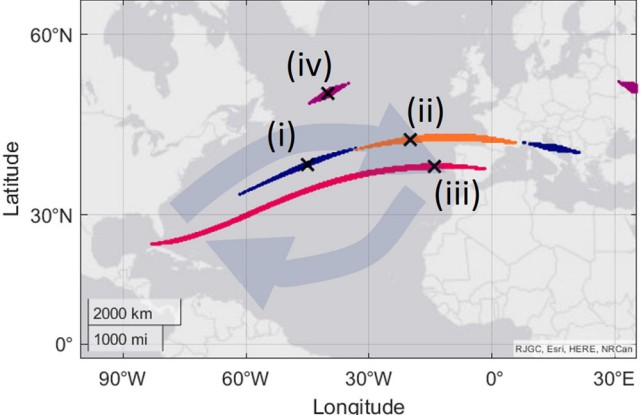

**Fig. 4 Possible locations for the virtual magnetic displacements in ref. [18].** Coloured areas show all possible locations with the same magnetic parameters as those used in the study. A different colour is used for each virtual magnetic displacement (i–iv). The large faded arrows indicate the flow direction of the Atlantic gyre.

the possible locations. In this case, it makes sense for the turtles to orient Eastwards at every one of the possible locations for (i) – this still aligns with the general direction of the gyre at these locations. For virtual displacement (ii), the possible locations (Fig. 4, orange dots) nearly overlap with those of (i). The difference between (i) and (ii) in magnetic parameters is 0.7° inclination and 500 nT total intensity. These may be on the edge of what is a perceptible difference, and yet turtles at (ii) orientated in a different direction from those at (i) – South-South East. Again, at each of the possible locations for (ii), the orientation does somewhat agree with the premise of attempting to stay within the gyre. For virtual displacement (iii) the possible locations span the entire Mid-Atlantic Ocean. The turtles at (iii) oriented South West. However, because the possible locations for this virtual magnetic displacement span such a wide range, it is impossible to make any meaningful interpretation of the orientation in relation to the gyre. This situation can make virtual magnetic displacement experiments problematic in light of realistic sensitivity values—when the interpretation relies upon the existence of a specific and unique location that does not exist. In the case of this study[18], this lack of clarity in interpretation does not damage the general finding that at different magnetic parameters turtles sometimes orient in differing directions. However, in other studies, there is a risk that the presence of alternate possible locations may relegate their core findings.

Another study on sea turtles found that they were able to perceive their longitudinal location with the use of a bi-coordinate magnetic map[19]. Two virtual magnetic displacements were performed, corresponding to two locations in the Atlantic Ocean that sit at the same latitude, but differing longitudes (Fig. 5). The observed difference in orientation direction at the virtual magnetic displacements was attributed to an ability to 'encode' longitudinal information from the two magnetic parameters. Looking at the possible locations for these displacements, it is immediately clear that many of the possible locations exist at different latitudes. This makes it impossible to state that it is longitudinal position alone that has been 'perceived' by the turtles. (Fig. 5, blue and pink areas). The human concepts of longitude and latitude do not need to be understood for effective navigation and focussing on them as a template for how animals navigate may be counterproductive. While it is true that the combinations of intensity and inclination used to signify these two virtual magnetic displacements do correspond to places

broadly on the east and west side of the Atlantic that are not overlapping, it is unlikely that the turtles have perceived unique locations as proposed. A closer look at displacement (ii) shows that the possible locations for the given magnetic parameters do not overlap with the suggested location for the displacement. It appears the total intensity values may have been rounded to the nearest 1000 nT in the original paper. This introduces a further important point, that any rounding of the parameter values will change the possible locations for the displacement. Limitations of the testing equipment can have a similar effect if the variance of the magnetic parameters that is achievable is higher than the realistic sensitivity of the animal. It is fair to conclude that the turtles have differing orientations at the two locations due to differences in magnetic parameters, but basing this upon specific locations on a map, and proposing that an inference has been made on the part of the turtles about their longitudinal position, is not supported by the data.

**Bi-coordinate maps worldwide**. To give a general picture of how multiple locations can appear on a world scale, and how it is important to consider multiple levels of sensitivity, we look at a virtual magnetic displacement study performed on salmon[20]. Five virtual magnetic displacements were performed, two in the Atlantic Ocean, two in the Pacific Ocean, and one under the natural conditions of the testing site on the East coast of the USA (Fig. 6a). Salmon from a non-migratory population were tested for orientation under the differing magnetic conditions (intensity and inclination) for each displacement. Using the pre-defined level of sensitivity to magnetic fields (±200 nT and ± 0.5°), we see that possible locations span wide areas, and reappear in multiple locations longitudinally (Fig. 6a, coloured areas). Of note, the possible locations corresponding to the magnetic parameters of the test site (blue area) almost exactly match those found at the ancestral site of the salmon (*, Fig. 6a). This limits the ability to discern between orientation directions with respect to the test site, or the ancestral site. A small decrease of the sensitivity values to ±0.75° inclination and ± 400 nT, significantly increases the possible locations such that some exist in both the Atlantic and Pacific Ocean for the same virtual displacements (yellow and pink areas, Fig. 6b). This makes comparisons between these two virtual magnetic displacements impossible because if an animal is sensitive to ±0.75° and ±400 nT, then based upon these two magnetic parameters alone, they will be unable to tell the difference between being located at the Southern Pacific virtual magnetic displacement site, or the any of the other possible locations (yellow areas, Fig. 6b) existing in the Mid Atlantic Ocean. This is reciprocally also true for the Southern Atlantic virtual displacement (pink areas, Fig. 6b). This highlights the importance of investigating a range of animal sensitivity estimates, especially when little is known about the sensitivity of the study species.

## Conclusions

In total, we have assessed 29 studies that performed virtual magnetic displacements. Six of the studies included declination as a possible magnetic cue[14,21–25]. As a result, thanks to the use of a tri-coordinate map, the possible locations for the virtual magnetic displacements in five of these papers were limited to the locations suggested. We, therefore, suggest that unless there is a priori reason to exclude declination, it should be included as a possible cue in virtual magnetic displacement studies. In one of the tri-coordinate studies[25], however, multiple locations existed for virtual magnetic displacements despite the use of declination (see Supplementary Fig. 9). It is important to note, therefore, that even in studies in which a tri-coordinate map is tested, a lack of knowledge of the animal's sensing ability means that problems

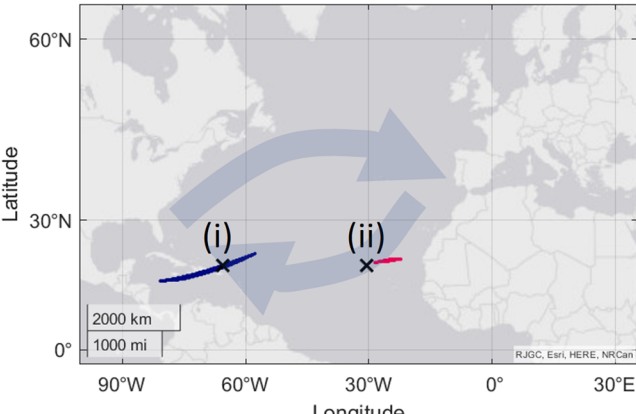

**Fig. 5 Possible locations for the two virtual magnetic displacements in ref. [19].** Coloured areas show all possible locations with the same magnetic parameters as those used in the study. A different colour is used for the two virtual magnetic displacements (i and ii). The large faded arrows indicate the flow direction of the Atlantic gyre.

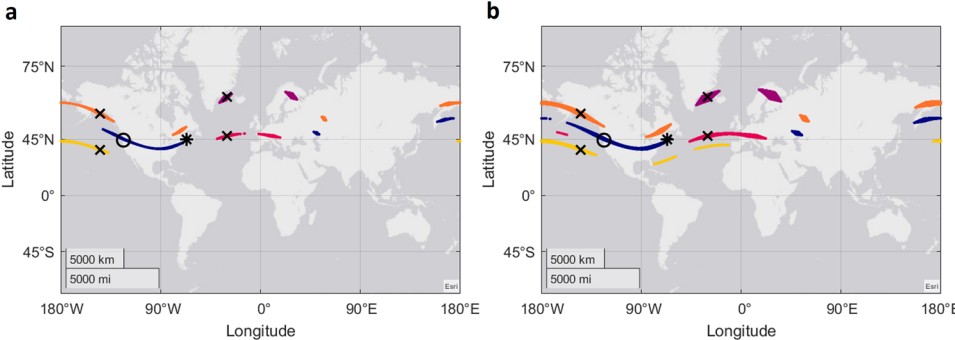

**Fig. 6 All possible virtual magnetic displacement locations from ref.** [20]. **a** Sensitivity set to ±0.5° inclination and ±200 nT total intensity. **b** The same virtual magnetic displacements but with sensitivity at ±0.75° inclination and ± 400 nT total intensity. Locations for each displacement as suggested in Scanlan et al., are denoted with a black cross, the possible locations for each are shown with coloured areas. The asterisk is the location of the ancestral site. Control test location under the ambient magnetic field is indicated with a black circle.

can still arise. Regardless of which cues are presented to an animal, it should not be assumed that the animal is utilising all of them. Therefore, all combinations of cues should be considered; how different combinations might change the possible locations, and in turn, how that might change the animal's likely perception of their location. Predictions that are key to the experimental design, and interpretations of the resulting data, should be robust to all combinations of cues.

Of the other 23 studies, all contained at least one virtual magnetic displacement that existed at multiple possible locations (Figs. 3–6 and Supplementary Figs. 1–8 and 10–20). These studies were performed on wide range of animals: birds[16,26–33], fish[20,34–38], reptiles[15,18,19,39–42], and crustaceans[2]. We have shown that at best, problems arising from multiple locations can introduce a small element of doubt to the interpretation of results and at worst, they can negate findings entirely.

We have not shown any examples of virtual magnetic displacement experiments from Asia or the Southern hemisphere as to our knowledge studies are lacking in these areas. Without them, we can only comment generally that the isolines of inclination and total intensity can diverge significantly from a linear pattern. This can potentially improve the viability of a bi-coordinate map, as the two magnetic parameters can confer distinctly different information, making for easily localisable unique positions. Perhaps more likely though, certainly over long distances, it can make localisation using only intensity and inclination more challenging, as inconsistent gradients of change and divergent relationships between parameters increase the complexity of usage. However, any judgements of the feasibility of a magnetic map in these areas need to be made with the specific locations, species, and ecological context in mind.

In light of our findings, virtual magnetic displacements remain a powerful tool in understanding magnetoreception in animals. For their potential to be fully realised, careful assessment of all possible virtual magnetic displacement locations, with considerations made for any lack of knowledge of sensitivity to magnetic parameters need to be made during both the design and evaluation phases. We provide our MATLAB application as an open-source method to make these assessments. It is possible that a route to a better understanding of animal sensitivity to magnetic parameters could be via titrated virtual magnetic displacements, whereby incremental changes to magnetic parameters may indicate sensitivity levels and behavioural responses as a function of the changed parameters. Furthermore, viewing magnetic parameters as context-specific indicators[43], or step-wise cues[44], rather than coordinates on a map, may yield further breakthroughs in understanding how animals sense and use magnetic fields. It is also paramount to keep in mind that magnetic parameter values,

and the relationships between them, fluctuate over time[45]. Finally, we also want to highlight that there may be a need for further investigation not only in the sensitivity of animals to magnetic fields, but in also which components of the field are detectable, and to what extent a detector may be sensitive to discrete $x$, $y$ and $z$ components. We, therefore, suggest that along with all multiple locations, the $x$, $y$, and $z$ components of the magnetic field for each virtual magnetic displacement, and the achieved standard deviations, are also routinely reported in scientific articles to aid with reproducibility.

## Methods

We developed a visualisation tool that allows for possible locations of virtual magnetic displacements to be viewed on a world map (ViMDAL). Magnetic parameters can be switched on or off, effectively changing which cues are being assumed as detectable. In the same way, the sensitivity to these cues can be changed, with the resulting possible locations either widening or reducing as sensitivity is decreased or increased. The date of the magnetic information can also be changed to match that of the study we are assessing. When using this tool, developed in MATLAB, to investigate the possible virtual magnetic displacement locations in published work, we have decided upon a sensitivity assumption of ±200 nT total intensity, and ± 0.5° inclination and declination. Whilst there is little published research into the possible sensitivity of animals to magnetic parameters, these assumptions are based upon the very highest sensitivity values that seem likely in magneto-receptive animals[46–49]. The resolution of the mapped locations is 0.25 degrees latitude/longitude. Magnetic values were calculated using the International Geomagnetic Reference Field (IGRF), 13th Generation.

We have investigated virtual magnetic displacement techniques that performed a 'full displacement', whereby magnetic components are intended to match those of another specific location. Other methods, such as changing only the $z$ component of the field are also used[50,51], but are not susceptible to the same issues as a full displacement, as there is no effort to map them to specific, real, or unique locations. We have also not investigated physical displacement studies, though we suggest that similar assessments as we have done here should also be performed when conducting a physical displacement. In other studies, only a single magnetic parameter is changed, such as inclination[17]. Whilst this is a reasonable method for investigating whether an animal is sensitive to a single magnetic parameter, the magnetic manipulation is not intended to map to any real locations elsewhere on Earth, and therefore is not included in this assessment. Similarly, we have not assessed other studies where magnetic parameters are changed in any other fashion that is not clearly linked to specific real-world locations[48]. Helmholtz coils are also commonly used to change the horizonal orientation of the magnetic field, for example, changing the direction of magnetic North. Again, we have not assessed these studies as they are focussed on understanding the magnetic compass, rather than map, and the changed magnetic parameters do not represent an attempt to model an alternative location.

**Reporting summary**. Further information on research design is available in the Nature Portfolio Reporting Summary linked to this article.

## Data availability

The datasets generated in the current study, and the code used to generate them, are available by downloading and running ViMDAL from the MathWorks file exchange

[https://uk.mathworks.com/matlabcentral/fileexchange/117795-vimdal]. The source data for the magnetic parameters shown in the figures were taken from each published study, and are also included as Supplementary Data 1.

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

## Acknowledgements

We thank John Phillips and two anonymous reviewers for helpful comments on the manuscript. This work was funded by a BBSRC grant (reference: BB/R001081/1) awarded to R.A.H., as well as a grant from the Leverhulme Trust (reference: RPG-2020-128) awarded to R.A.H. and to O.L. within the Sonderforschungsbereich (SFB) 1372 'Magnetoreception and Navigation in Vertebrates' (project-ID 395940726) by the Deutsche Forschungsgemeinschaft (DFG).

## Author contributions

W.S. developed the methodology, conceived the study, and wrote the initial manuscript. O.L., F.P., and R.H. contributed to the development of the study and the writing of the manuscript.

## Competing interests

The authors declare no competing interests. R.A.H. is an Editorial Board Member for *Communications Biology*, but was not involved in the editorial review of, nor the decision to publish this article.
