## [Peer Review File · Communications Biology]

Reviewers' comments:

Reviewer #1 (Remarks to the Author):

The authors make a valuable contribution by addressing the utility of a bicoordinate magnetic map, derived from spatial variation in the magnetic field, as a source of information about geographic position. I am particularly interested in this work because of my own research on the possibility of a 'short-range, high resolution' map in small terrestrial animals that move over distances of only a few kilometers (newts, mice, etc.).

There is general agreement in the literature that the 'sensor' that is responsible for measurements of spatial variation in the magnetic field is a magnetite-based mechanism. In addition to the magnetite-based 'map detector', many, and perhaps all, vertebrates appear to have a second light-dependent magnetoreception mechanism that functions as a compass. The quantum process underlying the light-dependent magnetic compass (i.e., radical pair mechanism) is generally assumed not to be capable of measuring tiny changes in the magnetic field necessary to derive map information.

One assumption made by the authors is that perception of spatial variation in the magnetic field by long-distance migrants corresponds to human characterization of the magnetic field as having X, Y, and Z components. Further consideration of this assumption may prove useful. Magnetite-based detectors that utilize an array of single domain and/or super-paramagnetic particles of magnetite are likely to be sensitive to the both the direction and intensity of the magnetic field. For this type of mechanism to selectively respond to specific components of the magnetic field requires that the particle array(s) consist of magnetite particles that are non-randomly aligned, so that their response to the intensity can be associated with a particular alignment relative to the magnetic field. One way to think about the differing effects of magnetic field direction and intensity on the response of individual magnetite particles is to consider an early model by Kirschvink and Gould in which the alignment of particles by the magnetic field was hypothesized to provide a measure of magnetic field direction (azimuth or inclination), while the variation around that average alignment (as the particle is jostled by Brownian motion) would provide a measure of magnetic field intensity; the greater the variation, the weaker the field. Consequently, the authors may want to point to the need for future research to determine how discrete X,Y and Z detectors would develop, and how they would be calibrated with the accuracy necessary to derive map coordinates.

As a basis for future discussions (not necessary to include in this paper), an interesting question is whether varying the authors' assumptions would provide additional insights into the possible designs of animal navigations systems and the reliance on magnetic input.

Using the type of mechanism described above to 'extract' a particular X,Y or Z component of the magnetic field would require the animal to take measurements with the particle array in a consistent alignment relative to the magnetic field, which would require an independent input to position the detector in a fixed alignment relative to the magnetic field. Our research suggests that newts do this using the light-dependent magnetic compass to align the magnetite-based map detector. As yet to be addressed in our work with newts (and presumably with other animal navigators) is that precise magnetic field measurements also require a gravity detector that is sensitive enough to align the magnetite-based detector in the vertical plane with the same resolution needed to align the magnetite-based map detector in the horizontal plane.

I wonder if there is an alternative to imposing human biases on how to characterize variation in the multiple components of the magnetic field. Imagine a more-or-less randomly aligned array of magnetite particles, each of which acts as an independent receptor. In this type of receptor, some combination of outputs from all the individual particle receptors would shift en masse when the intensity or inclination or declination of the magnetic field changes. Like the earlier example, this type of magnetoreception mechanism would require independent input(s) to position the particle array in

the vertical and horizontal planes.

It may be worth considering in future discussion whether this type of mechanism, or a combination of particle arrays with the particles aligned along different (although not necessarily orthogonal) axes, could operate as a 'null detector' that allows the animal to detect deviations from the baseline response of the entire particle array or arrays at a given target location. The task of identifying (or moving toward) a specific location --'labelled' by the hybrid output of this type of detector--might consist of moving in a direction that decreases the disparity between the current output of the receptor array and the 'null value' at the target site. The primary reason for considering the possibility of this type of detector is that it could underlie a 'magnetic map' that does not specify arbitrary locations on a bicoordinate grid, but only specifies deviations from the values at a target location. Instead of specifying a distinct location, the map would specify a direction of movement that decreases the disparity from the target value. This hypothesis shares some of the features of Kiepenheuer's (1984) proposal concerning the compass mechanism underlying the shifting migratory direction observed in some species of migratory birds.

One characteristic of the 'null detector' that might favor the evolution of this type of mechanism is that it might not require specific developmental mechanisms to produce and calibrate multiple particle arrays that respond to different components of the magnetic field.

When evaluating the possibility that migratory animals use something like the 'null detector' mechanism, the question would be what the path would look like for an animal starting at a relevant location, e.g., at the start of a seasonal migratory journey. Would the path resulting from the animal moving in a direction that decreases the disparity between the current output of the receptor and the null value at the target site converge on the target location or, instead, might there be a number of diverging paths, some of which would not reach the target area.

It is also worth considering whether other features of the terrain (shorelines, mountain ranges) might bias the choice between alternative paths, should they exist. Conversely, if this mechanism does specify more than one path, are there subsets of individuals from a given population that do in fact select alternative paths and end up in different target areas (i.e., different winter ranges for a migratory bird).

To step back for a moment, the reason for all the speculation is that long-distance migrations are unlikely to consist of movements from one arbitrary location to another on a bicoordinate grid. Many migratory songbirds, for example, use a 'vector strategy' during their first migration, flying in a genetically determined compass direction and distance. Only on the second migration do these migratory birds employ true navigation, i.e., relying on a map sense to return to the vicinity of their natal territory and, vice versa when returning in the second year to the same overwintering site. So, any mechanism that enables an individual to follow a path that will return it to a previously visited location is likely to be favored by natural selection, even if the return direction is not calculated from two bicoordinate map readings.

If further research provides support for something like the 'null detector' mechanism, consideration of the relative advantage and disadvantages of this mechanism, versus a 'simple' path integration and/or 'route reversal' mechanisms would be warranted.

Apologies for rambling on.
John Phillips

Reviewer #2 (Remarks to the Author):

This manuscript is not easy to review. The authors discuss experiments where animals are tested at one location in the magnetic field of a distant location and their directional response is recorded, but their comments concern mostly aspects that are self-evident and trivial. The authors do not distinguish between virtual magnetic displacements where the animals are familiar with the goal areas and the surrounding (magnetic) situation, and apparently spontaneous (innate?) responses to specific magnetic conditions (which are not directed towards a specific goal). It is true that the sensitivity of animals for magnetic directions and intensity is not known. What researchers observe in their test arenas, however, are directions with a certain, often considerable variance, but it is usually not clear what amount of this scatter is due to an unprecise concept of the goal, an unprecise directional response or a response to the artificial test situation. Because of this, it is completely unclear in what way virtual magnetic displacements should be a "powerful tool in understanding magnetoreception".

An interesting aspect is the mentioning of VIMDAL, but one would like to know more about it. Does it only show the main field, or are the local magnetic anomalies also included? How often is it to be updated?

Reviewer #3 (Remarks to the Author):

The MS 'Sense of doubt: Inaccurate and alternate locations of virtual magnetic displacements may give a distorted view of animal magnetoreception ability' by Will Schneider, Florian Packmor, Oliver Lindecke and Richard Holland reports the results of a very interesting study. The authors developed a tool for visualising virtual magnetic displacements and showed that some recent studies that utilised this technique were in fact ambiguous: they emulated magnetic conditions that existed not only in the intended location, but also in range of other locations. In some extreme cases, the given combination of magnetic parameters did not appear near the suggested location at all (Fuxjager et al., 2014) or did not exist anywhere on Earth (Keller et al., 2021). As the authors put mildly, it 'introduce[d] a small element of doubt to the interpretation of results' (line 325). This unfortunate situation occurred when the authors of the original studies recreated inclination and total field intensity of the intended displacement location, but disregarded declination. It is necessary to emphasize that the use of declination as a parameter of the presumed geomagnetic map is debated, and even the same group of researchers that first claimed that declination was a part of the positioning systems in migratory birds (Chernetsov et al., 2017) failed to reproduce this result in other species of birds (Chernetsov et al., 2020). However, the results reported in this study show very clearly that in some parts of the world, and particularly in the northern Atlantic, a magnetic map based on inclination and intensity alone (i.e. without declination) is hardly feasible. It makes the (yet unanswered) question of use of declination in this context even more important. It might be a matter of taste, but I feel that the MS would benefit from mentioning this point in the discussion.

Minor comments.

I assume that throughout figure legends in the Supplementary material, 'vertical magnetic displacements' should be 'virtual magnetic displacements'. I also suggest to make figure legends less uniform and more specific. Instead of writing 'Virtual magnetic displacement locations as suggested by the authors (if provided) are indicated with 'x's', I suggest to write 'Virtual magnetic displacement locations as suggested by the authors are indicated with 'x's' when they are provided, and 'Virtual magnetic displacement locations were not provided' when that was the case.

Reviewers' comments:

Reviewer #1 (Remarks to the Author):

The authors make a valuable contribution by addressing the utility of a bicoordinate magnetic map, derived from spatial variation in the magnetic field, as a source of information about geographic position. I am particularly interested in this work because of my own research on the possibility of a 'short-range, high resolution' map in small terrestrial animals that move over distances of only a few kilometers (newts, mice, etc.).

We are very glad that you find our manuscript interesting and to be a valuable contribution to the field.

There is general agreement in the literature that the 'sensor' that is responsible for measurements of spatial variation in the magnetic field is a magnetite-based mechanism. In addition to the magnetite-based 'map detector', many, and perhaps all, vertebrates appear to have a second light-dependent magnetoreception mechanism that functions as a compass. The quantum process underlying the light-dependent magnetic compass (i.e., radical pair mechanism) is generally assumed not to be capable of measuring tiny changes in the magnetic field necessary to derive map information.

One assumption made by the authors is that perception of spatial variation in the magnetic field by long-distance migrants corresponds to human characterization of the magnetic field as having X, Y, and Z components. Further consideration of this assumption may prove useful. Magnetite-based detectors that utilize an array of single domain and/or super-paramagnetic particles of magnetite are likely to be sensitive to the both the direction and intensity of the magnetic field. For this type of mechanism to selectively respond to specific components of the magnetic field requires that the particle array(s) consist of magnetite particles that are non-randomly aligned, so that their response to the intensity can be associated with a particular alignment relative to the magnetic field. One way to think about the differing effects of magnetic field direction and intensity on the response of individual magnetite particles is to consider an early model by Kirschvink and Gould in which the alignment of particles by the magnetic field was hypothesized to provide a measure of magnetic field direction (azimuth or inclination), while the variation around that average alignment (as the particle is jostled by Brownian motion) would provide a measure of magnetic field intensity; the greater the variation, the weaker the field. Consequently, the authors may want to point to the need for future research to determine how discrete X,Y and Z detectors would develop, and how they would be calibrated with the accuracy necessary to derive map coordinates.

Thank you for this comment. As you point out, we have focussed here on the X/Y/Z components of the field that are directly modelled within a Helmholtz coil, usually to match the magnetic parameters of inclination and total intensity found elsewhere. We agree entirely that there is much work to do to fully understand which components and parameters of the Earth's magnetic field are detected by animals, and that general assumptions are made without a huge amount of evidence. Indeed, here we seek to focus on the assumptions that are associated with the general methodology of performing virtual magnetic displacements, and hope that this work can draw attention to the areas in which we lack knowledge (sensitivity) and how assumptions may cause some research to be invalidated. We have added a comment regarding this to the manuscript [LINE 353].

As a basis for future discussions (not necessary to include in this paper), an interesting question is whether varying the authors' assumptions would provide additional insights into the possible designs of animal navigation systems and the reliance on magnetic input.

Using the type of mechanism described above to 'extract' a particular X,Y or Z component of the magnetic field would require the animal to take measurements with the particle array in a consistent alignment relative to the magnetic field, which would require an independent input to position the detector in a fixed alignment relative to the magnetic field. Our research suggests that newts do this using the light-dependent magnetic compass to align the magnetite-based map detector. As yet to be addressed in our work with newts (and presumably with other animal navigators) is that precise magnetic field measurements also require a gravity detector that is sensitive enough to align the magnetite-based detector in the vertical plane with the same resolution needed to align the magnetite-based map detector in the horizontal plane.

I wonder if there is an alternative to imposing human biases on how to characterize variation in the multiple components of the magnetic field. Imagine a more-or-less randomly aligned array of magnetite particles, each of which acts as an independent receptor. In this type of receptor, some combination of outputs from all the individual particle receptors would shift en masse when the intensity or inclination or declination of the magnetic field changes. Like the earlier example, this type of magnetoreception mechanism would require independent input(s) to position the particle array in the vertical and horizontal planes.

We strongly agree that removing the human bias, and the expectation that magnetic fields map to specific locations, would greatly advance our understanding of how animals can sense, and accordingly how they behave to a changed field. Sample size permitting, titration of magnetic parameters, in a similar fashion to your 2002 paper, incrementally in positive and negative directions in small steps, would begin to reveal the sensitivity level as well as behavioural responses as a function of the changing inputs (the magnetic parameters) [LINE 347]. In reverse to the standard methodology, this could then be mapped onto real locations to allow for interpretation of orientation behaviours in relation to how magnetic parameters map to geographic location. Clearly the challenge here is to build up a sufficient sample size, though this approach would surely progress our understanding significantly.

It may be worth considering in future discussion whether this type of mechanism, or a combination of particle arrays with the particles aligned along different (although not necessarily orthogonal) axes, could operate as a 'null detector' that allows the animal to detect deviations from the baseline response of the entire particle array or arrays at a given target location. The task of identifying (or moving toward) a specific location --'labelled' by the hybrid output of this type of detector--might consist of moving in a direction that decreases the disparity between the current output of the receptor array and the 'null value' at the target site. The primary reason for considering the possibility of this type of detector is that it could underlie a 'magnetic map' that does not specify arbitrary locations on a bicoordinate grid, but only specifies deviations from the values at a target location. Instead of specifying a distinct location, the map would specify a direction of movement that decreases the disparity from the target value. This hypothesis shares some of the features of Kiepenheuer's (1984) proposal concerning the compass mechanism underlying the shifting migratory direction observed in some species of migratory birds.

One characteristic of the 'null detector' that might favor the evolution of this type of mechanism is

that it might not require specific developmental mechanisms to produce and calibrate multiple particle arrays that respond to different components of the magnetic field.

When evaluating the possibility that migratory animals use something like the 'null detector' mechanism, the question would be what the path would look like for an animal starting at a relevant location, e.g., at the start of a seasonal migratory journey. Would the path resulting from the animal moving in a direction that decreases the disparity between the current output of the receptor and the null value at the target site converge on the target location or, instead, might there be a number of diverging paths, some of which would not reach the target area.

It is also worth considering whether other features of the terrain (shorelines, mountain ranges) might bias the choice between alternative paths, should they exist. Conversely, if this mechanism does specify more than one path, are there subsets of individuals from a given population that do in fact select alternative paths and end up in different target areas (i.e., different winter ranges for a migratory bird).

To step back for a moment, the reason for all the speculation is that long-distance migrations are unlikely to consist of movements from one arbitrary location to another on a bicoordinate grid. Many migratory songbirds, for example, use a 'vector strategy' during their first migration, flying in a genetically determined compass direction and distance. Only on the second migration do these migratory birds employ true navigation, i.e., relying on a map sense to return to the vicinity of their natal territory and, vice versa when returning in the second year to the same overwintering site. So, any mechanism that enables an individual to follow a path that will return it to a previously visited location is likely to be favored by natural selection, even if the return direction is not calculated from two bicoordinate map readings.

If further research provides support for something like the 'null detector' mechanism, consideration of the relative advantage and disadvantages of this mechanism, versus a 'simple' path integration and/or 'route reversal' mechanisms would be warranted.

The 'null detector' hypothesis certainly warrants a great degree of further exploration, and as you state here, its usage would likely be dependent upon the context within which an animal's navigation is framed, determining which strategy makes the navigational challenge possible or the most efficient. We are very happy that our manuscript has stimulated this fascinating discussion, and thank you very much for your thought provoking comments. We hope that it has the same effect on the whole field, and that further research can build upon these ideas.

Apologies for rambling on.
John Phillips

Reviewer #2 (Remarks to the Author):

This manuscript is not easy to review. The authors discuss experiments where animals are tested at one location in the magnetic field of a distant location and their directional response is recorded, but their comments concern mostly aspects that are self-evident and trivial.

We appreciate the comments of the reviewer but are sorry that they believe our findings to be self-evident and trivial. We disagree and feel that if the issues that we highlight were self-evident, then we would hope that they would be restricted to only a handful of studies, rather than concern almost all studies that we investigated (as was the case). We also believe that the highlighted issues are of a great importance to the field.

The authors do not distinguish between virtual magnetic displacements where the animals are familiar with the goal areas and the surrounding (magnetic) situation, and apparently spontaneous (innate?) responses to specific magnetic conditions (which are not directed towards a specific goal).

We agree that it is key to the planning, understanding, and interpretation of virtual magnetic displacement studies to distinguish between innate and learnt responses to magnetic fields. However, this does not alter the issues we highlighted – whether the hypothesis is that an animal has a learnt or innate response, the virtual presence of multiple locations for magnetic parameters can cause problems. We are wary of passing an in-depth judgement on the validity of each independent study, as this would require a great deal of discussion on the ecological context of each study species – and in which case, one would indeed have to consider whether or not the animal is predicted to have a learnt vs innate response. Instead, we feel the manuscript is stronger, more readable, and less likely to be accused of any bias, when we present the technical issues that we find and highlight when they may alter interpretation of results, rather than making those new interpretations ourselves.

It is true that the sensitivity of animals for magnetic directions and intensity is not known. What researchers observe in their test arenas, however, are directions with a certain, often considerable variance, but it is usually not clear what amount of this scatter is due to an unprecise concept of the goal, an unprecise directional response or a response to the artificial test situation. Because of this, it is completely unclear in what way virtual magnetic displacements should be a “powerful tool in understanding magnetoreception”.

All studies of animal magnetoreception have struggled with noise (Johnsen et al., 2020), and therefore whilst your concerns are very valid, they are not specific to the method of virtual magnetic displacements. We therefore stand by our statement that virtual magnetic displacements are a powerful tool in understanding magnetoreception. Despite the issues that we highlight, virtual magnetic displacements have enabled studies that would otherwise be impossible, and conducted properly have produced remarkable results (e.g. Kishkinev et al, 2015). At this point in time, i.e., after several papers have used virtual magnetic displacements, the issues we identified can be pointed out, and based on our description and or the VIMDAL tool we provide, experiments can be refined with a chance of future studies being less ambiguous. The “noise” or scatter introduced and shown by some study models (mostly vertebrates) are compensated for by rigorous pre-screenings of behavioural phenotypes tested, or parallel investigations of the genetic make-up of the respective specimen (e.g. Delmore & Irwin 2014, <https://doi.org/10.1111/ele.12326>). However, these methods vary between species, study populations etc., and descriptions are therefore part of the specific studies. In any case, these are “biological issues” whilst we are focusing on the physical aspects of virtual magnetic displacements. We, in fact, expect a reduction of this “biological noise” as a consequence of the suggested methodological refinements.

An interesting aspect is the mentioning of VIMDAL, but one would like to know more about it. Does it only show the main field, or are the local magnetic anomalies also included? How often is it to be updated?

VIMDAL uses the IGRF (International Geomagnetic Reference Field) for the magnetic parameters. This model gives the main field rather than local magnetic anomalies (though we thank you for your comment as these are also of interest and would be a worth exploring in the future). The IGRF is updated every 5 years, covering the next 5 year period. Downloading the new IGRF model into VIMDAL will update the source data, and we will be sure to provide documentation on this process when the new model is released.

Reviewer #3 (Remarks to the Author):

The MS 'Sense of doubt: Inaccurate and alternate locations of virtual magnetic displacements may give a distorted view of animal magnetoreception ability' by Will Schneider, Florian Packmor, Oliver Lindecke and Richard Holland reports the results of a very interesting study. The authors developed a tool for visualising virtual magnetic displacements and showed that some recent studies that utilised this technique were in fact ambiguous: they emulated magnetic conditions that existed not only in the intended location, but also in range of other locations. In some extreme cases, the given combination of magnetic parameters did not appear near the suggested location at all (Fuxjager et al., 2014) or did not exist anywhere on Earth (Keller et al., 2021). As the authors put mildly, it 'introduce[d] a small element of doubt to the interpretation of results' (line 325). This unfortunate situation occurred when the authors of the original studies recreated inclination and total field intensity of the intended displacement location, but disregarded declination. It is necessary to emphasize that the use of declination as a parameter of the presumed geomagnetic map is debated, and even the same group of researchers that first claimed that declination was a part of the positioning systems in migratory birds (Chernetsov et al., 2017) failed to reproduce this result in other species of birds (Chernetsov et al., 2020). However, the results reported in this study show very clearly that in some parts of the world, and particularly in the northern Atlantic, a magnetic map based on inclination and intensity alone (i.e. without declination) is hardly feasible. It makes the (yet unanswered) question of use of declination in this context even more important. It might be a matter of taste, but I feel that the MS would benefit from mentioning this point in the discussion.

We thank the reviewer for their comments. We agree that excluding declination as a possible cue reduces the possible localisation potential of a magnetic map, though it is also important to note that including it does not necessarily mean that the test animal will benefit. The design of these studies should consider that the test animal may use all of these cues (inclination, intensity, and declination), perhaps only one, or any combination of pairs. Predictions and interpretations of data should therefore be robust to how the inclusion of more or fewer of the magnetic parameters included can alter the possible locations where these values are found. We have added lines to discussion to clarify these important points [STARTING LINE 315].

We also urge caution with the interpretation of Chernetsov et al. (2020). As it stands the study does not provide specific evidence that declination is not used by these species. Because they created a virtual magnetic location in Scotland that required only the changing of declination, the design lacks a positive control to show whether the other two parameters (intensity and inclination) are used if they are changed (see for example Kishkinev, Packmor et al. 2021). At this stage, the study provides evidence that robins and garden warblers do not respond to a magnetic displacement in which the change in parameters matched a location far to the west of their migratory route, but we cannot be certain whether this is because they don't use declination, don't use the magnetic field for navigation at all, or reject it when in conflict with other cues, or

for some other non-navigational reason why they have not responded to the virtual displacement.

Minor comments.

I assume that throughout figure legends in the Supplementary material, 'vertical magnetic displacements' should be 'virtual magnetic displacements'. I also suggest to make figure legends less uniform and more specific. Instead of writing 'Virtual magnetic displacement locations as suggested by the authors (if provided) are indicated with 'x's', I suggest to write 'Virtual magnetic displacement locations as suggested by the authors are indicated with 'x's' when they are provided, and 'Virtual magnetic displacement locations were not provided' when that was the case.

Thank you for noticing this error, and we have updated the figure legends as per your suggestion.

REVIEWERS' COMMENTS:

Reviewer #1 (Remarks to the Author):

The authors provide a comprehensive discussion of the pros and cons of different approaches to investigating the possibility of a magnetic map, and an excellent public access software package to locate specific values of potential map components anywhere on the earth's surface. Earlier reviewers comments have been adequately addressed. The authors' contribution should prove useful for investigators working in this area of research.

Reviewer #2 (Remarks to the Author):

The paper is improved.

However, the authors still fail to discuss thoroughly the different consequences of simulating magnetic sites for animals that head to a familiar location, i.e., where the simulated site is the starting point of a navigational process towards a specific goal, and innate responses to specific magnetic conditions, i.e., where the animals just want to leave the site heading in an innate direction.

It would be nice to discuss this in some detail.

Reviewer #3 (Remarks to the Author):

I am happy with how that authors responded to my earlier comments. I think that they also have very constructively responded to the comments made by Reviewer 1, John Phillips. And I would like to mention that I disagree with Reviewer 2 in their opinion that authors' 'comments concern mostly aspects that are self-evident and trivial'. No, as someone working in this particular field and having co-authored several of the papers discussed by the authors, I do not find their comments self-evident. Not at all. I see this paper as a valuable contribution to the field.

I also disagree with the notion by Reviewer 2 that due to the scatter in their preferred directions shown by animals tested in round arenas, 'it is completely unclear in what way virtual magnetic displacements should be a "powerful tool in understanding magnetoreception"'. Scatter is inevitable when we test orientation preferences of animals in a lab setting. Yes, it is a weakness of this method. However, at least 80% of what we have learned about long-distance orientation and navigation of animals since Gustav Kramer (1949: *Über Richtungstendenzen bei der nächtlichen Zugunruhe gekäfigter Vögel*. *Ornithologie als Biologische Wissenschaft* (Carl Winter Verlag), pp. 269–283) we know thanks to tests in round arenas. The method is not perfect, but neither is our world in general.

REVIEWERS' COMMENTS:

Reviewer #1 (Remarks to the Author):

The authors provide a comprehensive discussion of the pros and cons of different approaches to investigating the possibility of a magnetic map, and an excellent public access software package to locate specific values of potential map components anywhere on the earth's surface. Earlier reviewers comments have been adequately addressed. The authors' contribution should prove useful for investigators working in this area of research.

We thank the reviewer for their positive feedback.

Reviewer #2 (Remarks to the Author):

The paper is improved.

However, the authors still fail to discuss thoroughly the different consequences of simulating magnetic sites for animals that head to a familiar location, i.e. where the simulated site is the starting point of a navigational process towards a specific goal, and innate responses to specific magnetic conditions, i.e. where the animals just want to leave the site heading in an innate direction.

It would be nice to discuss this in some detail.

We thank the reviewer for their comments, though we disagree with the need to discuss further the difference between a learnt response (animals homing to a known location), and innate responses an animal may have to magnetic fields. We do agree that this difference is incredibly important in understanding the ways in which animals can use the Earth's magnetic field; it is important in both the design of experiments, and in the interpretation of their results. However, it is a quite separate issue from the one that we present in this paper. The problems with multiple locations exist regardless of the way an animal is using the Earth's magnetic field. In some cases these problems may cause serious problems, in others they may not – perhaps because the existence of other possible locations appear on the other side of the globe where there is little chance of the animal confusing them (though we would argue the presence of these other locations should still be reported). It may be confusing to readers to reference differences between learnt responses and innate responses, when they may both be equally affected (or not) by the presence of multiple locations. Therefore we believe that the paper is stronger and clearer without attempting to introduce this topic.

Reviewer #3 (Remarks to the Author):

I am happy with how that authors responded to my earlier comments. I think that they also have very constructively responded to the comments made by Reviewer 1, John Phillips. And I would like to mention that I disagree with Reviewer 2 in their opinion that authors' 'comments concern mostly aspects that are self-evident and trivial'. No, as someone working in this particular field and having co-authored several of the papers discussed by the authors, I do not find their comments self-evident. Not at all. I see this paper as a valuable contribution to the field.

I also disagree with the notion by Reviewer 2 that due to the scatter in their preferred directions shown by animals tested in round arenas, 'it is completely unclear in what way virtual magnetic

displacements should be a “powerful tool in understanding magnetoreception”. Scatter is inevitable when we test orientation preferences of animals in a lab setting. Yes, it is a weakness of this method. However, at least 80% of what we have learned about long-distance orientation and navigation of animals since Gustav Kramer (1949: Über Richtungstendenzen bei der nächtlichen Zugunruhe gekäfigter Vögel. *Ornithologie als Biologische Wissenschaft* (Carl Winter Verlag), pp. 269–283) we know thanks to tests in round arenas. The method is not perfect, but neither is our world in general.

We thank the reviewer for their positive feedback and agree with their comments RE the use of virtual magnetic displacements as a tool for understanding magnetoreception.